# Whole-Exome and Transcriptome Sequencing Expands the Genotype of Majewski Osteodysplastic Primordial Dwarfism Type II

**DOI:** 10.3390/ijms241512291

**Published:** 2023-07-31

**Authors:** Flaviana Marzano, Matteo Chiara, Arianna Consiglio, Gabriele D’Amato, Mattia Gentile, Valentina Mirabelli, Maria Piane, Camilla Savio, Marco Fabiani, Domenica D’Elia, Elisabetta Sbisà, Gioacchino Scarano, Fortunato Lonardo, Apollonia Tullo, Graziano Pesole, Maria Felicia Faienza

**Affiliations:** 1Institute of Biomembranes, Bioenergetics and Molecular Biotechnologies, IBIOM–CNR, 70126 Bari, Italy; f.marzano@ibiom.cnr.it (F.M.); a.tullo@ibiom.cnr.it (A.T.); 2Department of Biosciences, University of Milan, 20133 Milan, Italy; matteo.chiara@unimi.it; 3Institute for Biomedical Technologies, ITB-CNR, 70126 Bari, Italy; arianna.consiglio@ba.itb.cnr.it (A.C.); valentina.mirabelli@gmail.com (V.M.); domenica.delia@ba.itb.cnr.it (D.D.); elisabetta.sbisa@ba.itb.cnr.it (E.S.); 4Neonatal Intensive Care Unit, Di Venere Hospital, 70012 Bari, Italy; 5Medical Genetics Unit, ASL Bari, 70012 Bari, Italy; mattia.gentile@asl.bari.it; 6Department of Clinical and Molecular Medicine, Sapienza University, 00185 Rome, Italy; maria.piane@uniroma1.it; 7Sant’Andrea University Hospital, 00185 Rome, Italy; camilla.savio@gmail.com; 8Department of Experimental Medicine, Sapienza University of Rome, 00185 Rome, Italy; marco.fab88@gmail.com; 9Medical Genetics Unit, AORN “San Pio”, Hosp. “G. Rummo”, 82100 Benevento, Italy; gioac.scarano51@gmail.com (G.S.); fortunato.lonardo@ao-rummo.it (F.L.); 10Department of Biosciences, Biotechnology and Biofarmaceutics, University of Bari “Aldo Moro”, 70126 Bari, Italy; 11Pediatric Section, Department of Precision and Regenerative Medicine and Ionian Area, University “A. Moro” of Bari, 70124 Bari, Italy

**Keywords:** ES, RNA-Seq, MOPDII, Majewski, pathogenic variants

## Abstract

Microcephalic Osteodysplastic Primordial Dwarfism type II (MOPDII) represents the most common form of primordial dwarfism. MOPD clinical features include severe prenatal and postnatal growth retardation, postnatal severe microcephaly, hypotonia, and an increased risk for cerebrovascular disease and insulin resistance. Autosomal recessive biallelic loss-of-function genomic variants in the centrosomal pericentrin (PCNT) gene on chromosome 21q22 cause MOPDII. Over the past decade, exome sequencing (ES) and massive RNA sequencing have been effectively employed for both the discovery of novel disease genes and to expand the genotypes of well-known diseases. In this paper we report the results both the RNA sequencing and ES of three patients affected by MOPDII with the aim of exploring whether differentially expressed genes and previously uncharacterized gene variants, in addition to *PCNT* pathogenic variants, could be associated with the complex phenotype of this disease. We discovered a downregulation of key factors involved in growth, such as IGF1R, IGF2R, and RAF1, in all three investigated patients. Moreover, ES identified a shortlist of genes associated with deleterious, rare variants in MOPDII patients. Our results suggest that Next Generation Sequencing (NGS) technologies can be successfully applied for the molecular characterization of the complex genotypic background of MOPDII.

## 1. Introduction

Majewski Osteodysplastic Microcephalic Primordial Dwarfism type II (MOPDII; OMIM #210720) is a rare, autosomal recessive skeletal dysplasia characterized by severe prenatal and postnatal growth retardation, severe postnatal microcephaly, abnormal dentition, skeletal anomalies, and increased risk for cerebrovascular disease and insulin resistance [1,2,3]. The majority of MOPD II cases are associated with biallelic loss-of-function of genomic variants in the centrosomal pericentrin (*PCNT*) gene on chromosome 21q22 [4,5]. *PCNT* is an integral component of the centrosome and has important roles in various indispensable cellular processes, such as microtubule organization, cell division, cell cycle progression, and assembly of cilia, working as a multifunctional scaffold for the assembly of different proteins [6]. Accordingly, in MOPDII subjects with a dysfunctional *PCNT*, cellular proliferation is probably impaired, explaining a reduction in the number of cells and the substantial growth failure observed in these patients [7].

Over the past decade, the development of the Next Generation Sequencing (NGS) technologies has resulted in the widespread usage of large scale “omics” assays in clinical practice. Exome sequencing (ES) is one of the most popular NGS applications for diagnostics and clinical screening [8]. ES has been successfully employed both for the discovery of novel disease genes, associated with pathological conditions [9], and expanding the genotypes of well-known diseases [10].

Recent studies [11] have demonstrated that, in addition to the delineation of genetic profiles of affected individuals, the characterization of the transcriptome profile by massive RNA sequencing (RNA-Seq) can be even more important for precision medicine applications, since it can capture a higher level of complexity, thus helping us to better understand the molecular mechanisms that underlie complex pathologies.

To date, no large-scale analysis of MOPDII disease cases has been reported in the literature. We performed the RNA-Seq and ES of three patients affected by MOPDII. The aim of this study was to explore whether differential gene expression and/or previously uncharacterized gene variants, in addition to *PCNT* pathogenic variants, could explain the multifaceted clinical symptoms of this disease. Our analysis showed, in all three patients, a dramatic downregulation of the key factors involved in prenatal and postnatal growth, such as IGF1R, IGF2R, and RAF1, and, in two cases, also CREBBP.

Furthermore, ES allowed us to identify heterozygous variants in several genes that could be implicated in some clinical features of MOPDII.

## 2. Results

### 2.1. Array CGH and Exome Sequencing

Array CGH was performed in addition to the conventional karyotype in order to determine if small, submicroscopic genomic deletions and/or duplications (1 kb to 10 Mb) were present in Patient 1. No significant unbalanced rearrangements were found.

To investigate the genetic background of MOPDII cases, we performed ES of Patient 1 (age 8 years) and his healthy parents. Afterwards, we extended the ES to the other two MOPDII patients (cases 2–3).

Coverage statistics (Appendix A) suggest that more than 95% of the target regions display a coverage of 20X or greater in all the samples analyzed, in line with the recommendations for the applications of targeted resequencing approaches in clinical studies [12].

A total of 171,352 genetic variants were identified in the exomes of the five individuals included in our study (three patients and the two parents of one of them). Among these, 170,962 were already included in the dbSNP database (build 151) or in other publicly available resources of human genetic variants. A total of 94,918 genetic variants were in exons of Refseq gene models (release 106) of the human hg19 genome assembly, while 76,434 variants were associated with introns. Of note 97.96% of intronic variants were found within a distance of 100 bp from a splicing donor or acceptor site, suggesting that, consistent with previous reports [13], the vast majority of intronic variants identified by ES lies in the vicinity of exon–intron boundaries.

Among the genetic variants associated with exons, 49,196 (51.83%) were in untranslated regions (3′ or 5′ UTR), 38,422 (40.48%) were in protein-coding exons, and 7300 (7.6%) were associated with exons of long non-coding RNA genes. A similar proportion of synonymous and nonsynonymous substitutions was observed in protein-coding genes: 49.9% and 46.15% of the total number of variants, respectively.

A total of 623 (0.36%) variants were predicted to have a potentially disruptive effect on a gene, and 233 of these (37.4%) were in the proximity of a splice site, while 390 (62.6%) were associated with the insertion of a premature stop codon, or with frameshifts in the protein-coding sequence.

A total of 172 and 150 variants had a minor allele frequency (MAF) ≤ 1 × 10^−5^, were homozygous in Patient 3 and Patient 2, respectively, and not homozygous in the healthy parents of Patient 1 (New Appendix A). The equivalent figure of Patient 1 is of only eight variants, suggesting that approaches based on the sequencing of complete trios [14] are highly effective for filtration of candidate genomic variants. When only the genetic variants associated with a radical effect on the gene product were considered (frameshift insertions/deletions, splice site variants, stop-gain/stop-loss variants, deleterious non synonymous substitutions), the number of candidate disease-causing variants was reduced to three for Patient 3, one for Patient 1, and five for Patient 2 (Table 1).

No de novo disease-causing variants or compound heterozygous variants were identified in Patient 1, for whom genetic profiles of the parents were also available. On the contrary, rare, potentially deleterious compound heterozygous variants, compatible with the incidence of MOPDII, were detected in Patient 2 and Patient 3 (* in Table 1)at the ZNF846 and DST genes, respectively (Table 1).

Importantly, this reduced list of candidate variants contained both the disease-causing variants in the *PCNT* gene that were previously characterized by Sanger sequencing in Patients 1 and 2 [4,7].

The three candidate disease-causing variants in Patient 3 were not subjected to further analyses since defects in these genes (Table 1) were considered unlikely to underlie the severe phenotype associated with MOPDII (Appendix A). We noticed that between 6.76% (Patient 3) and 14.08% (Patient 1) of the functionally relevant regions (here defined as the complete set of the Refseq exons plus 50 bp upstream and downstream) of the *PCNT* gene are covered by less than 10 reads, suggesting that our observations with respect to the absence of potentially disease-causing variants in Patient 3, based on ES alone, cannot be considered conclusive. However, a targeted Sanger resequencing of PCNT exons in the same patient confirms that there are no uncovered candidate disease-causing variants. Patient 3, despite the lack of known or novel candidate pathogenic variants in the PCNT gene, lacks the protein (Figure 1). The RNA-Seq analysis shows that PCNT transcript reads cover a high number of exons (29 of 48 exons) and, importantly, all the last exons, thus indicating that the entire gene is transcribed.

The absence of PCNT protein needs further investigations.

To identify possible genetic determinants of the comorbidities associated with MOPDII, we performed variant filtration and identified genes associated with rare, disruptive genomic variants in two or more patients. A total of 82 variants were retrieved by our filters (AF <= 1 × 10^−4^, deleterious effect on the gene function, not homozygous in the healthy parents of Patient1) (new Appendix A); four genes carried disruptive low-frequency variants in two or more patients (Table 2).

All of these genes have an important role in fetal development: RAD17 is essential for sustained cell growth; NPRL3 has a role in neuronal development; BCLAF1 (also reported in Table 1 because it mutated in Patient 3) and PRIM2, respectively, have a function as a transcriptional repressor in the first and as a factor of DNA synthesis in the second, playing essential roles in the development of the placenta.

### 2.2. Transcriptome Sequencing

Despite the extraordinary impact of whole-exome sequencing (ES) on the molecular genetics of Mendelian disorders, over 50% of the patients do not receive a genetic diagnosis after ES. This is probably due to the lack of a detailed functional annotation for synonymous or non-coding variants. Many of these variants might affect the expression levels of isoform abundances. Furthermore, ES regions cover only about 2% of the genome. For these reasons, in addition to the ES analysis, we also performed a whole transcriptome analysis of blood samples from Patient 1, and two healthy age- and sex-matched controls. The analysis showed 56 up-regulated and 138 down-regulated genes, for a total of 194 genes with significant changes in their level of expression (Appendix A).

The functional enrichment analysis of differentially expressed genes performed with DAVID showed the enrichment of these genes in diverse molecular functions, biological processes, and pathways (Appendix A). The most relevant result to our aims is the statistical enrichment of the genes in the “Insulin-like growth factor-activated receptor activity” (*p*-value = 0.03). Two key genes belonging to this category are IGF1R and IGF2R, which were both down-regulated in Patient 1 compared to controls (log2FC −1.43 and −0.58, respectively, and adjusted *p*-value < 0.01). This result is particularly relevant considering that the growth hormone (GH)–Insulin-Like Growth Factors (IGFs)–Insulin-Like Growth Factors Binding Protein 3 (IGFBP3) axis is a key endocrine modulator of prenatal and postnatal growth and metabolism, confirming an alteration of growth pathways. [15,16].

Next, we investigated whether other genes belonging to the IGF1R and IGF2R pathway were also differentially expressed in Patient 1. To this end, we used GenesLikeMe, which calculated similarity scores between IGF1R or IGF2R and all remaining candidate genes in the GeneCards database based on the SuperPaths attribute. Using the GenesLikeMe database, we selected 100 correlated genes for IGF1R and IGF2R. By comparing them, we identified 62 common genes. We searched for these 62 genes related to IGF1R and IGF2R in our list of 194 differentially expressed genes obtained by the RNA-Seq experiments. Interestingly we found that other two key genes involved in cell cycle division, apoptosis, cell differentiation, and cell migration, namely RAF1 and CREBBP, were also significantly down-regulated (log2FC -0.79 and -0.73 respectively, and adjusted *p*-value < 0.01; HYPERLINK “http://david.ncifcrf.gov” (accessed on 15 November 2022), HYPERLINK https://glm.genecards.org (accessed on 21 November 2022)).

The downregulation of the IGF1R, IGF2R, RAF1, and CREBBP genes was confirmed by RT-qPCR experiments. Interestingly, we also found that the other two patients showed a significant downregulation of IGF1R, IGF2R, and RAF, while CREBBP downregulation was confirmed in two patients out of three compared to age- and sex-matched controls (Figure 2).

## 3. Discussion

Microcephalic primordial dwarfisms (MPDs) are a group of autosomal recessive disorders characterized by an extreme growth failure which starts early in the development and continues postnatally, with a final height often reduced to 1 m [17].

Many distinct forms of MPD are recognized, clinically or through molecular diagnosis. MOPD I and MOPD III have been considered variants of the same disorder, previously described as cephaloskeletal dysplasia [18].

MOPDII, the most common and well-described type of microcephalic dwarfism, in addition to the classical features of the other MPD forms (severe prenatal and postnatal growth retardation and marked microcephaly), shows other clinical and radiographic characteristics [19]. These include skeletal dysplasia as long thin bones, small iliac wings with flat acetabular angles, dislocation or subluxation of the radial heads and hips, epiphyseal ossification delay, mesomelia, scoliosis (particularly in girls in late childhood or at puberty), and abnormal dentition. A major cause of death for MOPDII patients is neurovascular complications arising from both brain aneurysms and arterial narrowing, which results in multiple fragile collateral blood vessels (moyamoya). This condition may cause ministrokes (transient ischemic attack), stroke, or bleeding in the brain. Another important clinical condition not attributable to a dysfunctional *PCNT* is that most patients with MOPDII develop insulin resistance leading to skin pigmentation (acanthosis nigricans) and type II diabetes mellitus [20].

Against a very complex and dramatic clinical picture, at present, at the molecular level, the diagnosis of MOPDII is confirmed by the presence of biallelic loss-of-function pathogenic variants in the *PCNT* gene, which encodes a core centrosomal protein that, as a major constituent of the pericentriolar material, facilitates the nucleation of the mitotic spindle [21].Therefore, it is likely that in cells with mutated *PCNT,* the cellular proliferation is damaged, leading to a reduction in cell number and a substantial growth failure.

To date, there have been no large-scale studies on MOPDII disease. NGS technology has recently emerged as a useful alternative for determining genetic variants and alterations in the transcriptome, contributing to monogenic and polygenic disease pathogenesis.

In this study, for the first time we performed the exome sequencing of three MOPDII patients. We confirmed the pathogenic variants in two out of three patients in the *PCNT* gene, while the third patient, despite the lack of known or novel candidate pathogenic variants in the *PCNT* gene, lacked the protein [4,5,6,7].

Several different reasons could explain these findings. Pathogenic genomic variants might be associated with genomic regions not assayed by ES sequencing, such as introns, promoters, or enhancers, which are only detectable with whole genome sequencing.

Alternatively, post-transcriptional mechanisms or incorrect targeting by miRNAs could prevent the production of the protein. Additional investigations will be required to uncover/identify the genomic determinants of MOPDII in Patient 3.

In addition, ES provided an opportunity to research the genetic variants that are possibly associated with the highly variegated phenotype associated with MOPDII.

The Rad17-replication factor C (Rad17-RFC) and Rad9-Rad1-Hus1 complexes are thought to function in the early phase of cell-cycle checkpoint control as sensors for genome damage and genome replication errors. However, genetic analysis of the functions of these complexes in vertebrates is complicated by the lethality of these gene disruptions in embryonic mouse cells [22].

NPRL3 protein has a role in cortical development. People with MOPDII have an adult brain size comparable to that of a 3-month-old infant and intellectual development may be normal or with retardation [23]. The other two hypermutated genes in MOPDII patients are expressed in placenta during embryonic growth. In fact, placental BCLAF1 is an epigenetically regulated gene with random monoallelic expression in human placenta, and it participates in the regulation of cellular apoptosis and tissue development with other genes [24]. PRIM2 expression has been detected in several tissues, including brain, liver, blood, and placenta, and has been reported as an imprinted, maternally expressed gene in human white blood cells (WBCs). Its role in placental function is not well understood [25].

In the literature, we did not find pathogenic variants in genes belonging to the GH–IGFs–IGFBP3 axis, which are key modulators for the growth of the whole organism, as mentioned before.

One approach to enhance the evaluation of genetic variants could be the integration with functional genomic information deriving from the RNA-Seq, which provides direct insights into transcriptional perturbations caused by genetic changes in regulatory regions or in any step of gene expression, including non-coding RNAs. This was the approach we used and, indeed, in addition to the results obtained with ES analysis, the transcriptomic analysis also provided even more interesting outcomes. Specifically, the RNA-Seq analyses showed a downregulation of *IGF1R*, *IGF2R,* and *RAF1* in all three patients and *CREBBP* in two out of three subjects. The *IGF1R* gene plays an important role in prenatal and postnatal growth. Indeed, it is reported that IGF1R haplo-insufficiency leads to severe intrauterine and postnatal growth retardation and other delayed motor and mental development [26,27,28]. This finding is particularly relevant if we consider that Patient 1 showed a low IGF-1 serum level, and neither intensive nutrition, growth hormone, or IGF-1 intervention influenced growth outcome [7].

RAF1 and CREBBP have a central role in signal transduction pathways during development. Notably, pathogenic variants in *RAF1* are associated with Noonan syndrome, a genetic disorder characterized by short height, congenital heart disease, bleeding problems, and skeletal malformations [29]. Pathogenic variants in CREBBP cause Rubinstein–Taybi syndrome (RTS), a rare genetic disease characterized by short stature, mental retardation, and increased risk of developing solid malignancies and leukemia [30].

In conclusion, the transcriptomic and exome sequencing of three patients affected by MOPDII allowed us to uncover the differential expression of key genes involved in the growth process, and different non-synonymous variants in genes, whose pathogenic variants cause clinical symptoms and features that are strongly suggestive of MOPDII disease. Altogether, these results provide evidence that the genotypic background of this disease seems more complex than previously believed.

## 4. Materials and Methods

### 4.1. Subjects

Three children affected by MOPDII and the healthy parents of Patient 1 were analyzed with ES experiments. Three children, matched for age and sex with MOPDII patients, were used as controls in the RNA-Seq and RT-qPCR experiments.

All the procedures used were in accordance with the guidelines of the Declaration of Helsinki in 1995 (as revised in Seoul 2008) on Human Experimentation. The study protocol was approved by the Independent Ethics committee, Azienda Ospedaliero-Universitaria “Consorziale Policlinico” of Bari, Italy (protocol number 0076954). Informed consent was obtained from the parents of participating children.

**Patient 1** is a Caucasian male, born prematurely with a severe intrauterine growth retardation (IUGR) from non-consanguineous healthy parents. At birth, weight was −3.9 standard deviation score -SDS, length −4.7 SDS, head circumference −3.8 SDS. He showed facial dysmorphisms with a prominent beaked nose, ocular protrusion, micrognathia, absence of the earlobe, fine and sparse hair, fifth finger clinodactyly, and micropenis. Brain magnetic resonance imaging (MRI) showed mild cortical thickening, thinning of the corpus callosum, and delayed myelination. Subsequently, molecular analysis of the *PCNT* gene showed a homozygous splicing site pathogenic variant in position c.3608-2 A > G of intron 18. The variant was present in heterozygous state in both parents. At our first observation, he showed extremely short stature (−10.3 SDS), weight (−22.1 SDS), head circumference (−8 SDS), and severe bone age retardation. He also showed facial dysmorphisms, small dysplastic teeth, and short limbs with relatively short forearms [7].

**Patient 2** is a Caucasian male, born prematurely from non-consanguineous healthy parents. Birth weight was −2.33 SDS, birth length −1.95 SDS, head circumference −1.89 SDS. Postnatally, he was diagnosed with vescico-ureteral reflux with arterial hypertension. Severe growth delay was noted and a standard cytogenetic analysis was performed (46,XY). At the time of our first observation, he presented short stature −10.1 SDS, −9.4 SDS, and severe microcephaly, −3.4 SDS, growth failure typical of the syndrome. Dysmorphic features included high forehead with receded hairline, sparse scalp hair, convex nasal ridge, and mild retrognathia. The radiographs showed metaphyseal widening, absence of the ossification nuclei in the femoral head, and irregular distal femoral epiphyses. A year after our first observation, the patient developed paresis of the right arm as a consequence of a stenosis of the median cerebral artery [4].

His voice was squeaky and the teeth were small. X-ray examination showed high iliac wings, narrow ischia and pubis, overtubulated long bones, delta-shaped distal femoral metaphysis, and marked widening brachytelemesophalangia with delayed bone age [4]. Based on the clinical spectrum, an alternative diagnosis for MOPDII syndrome was supposed, confirmed by the subsequent identification of a homozygous single base insertion (c.1527_1528insA) in exon 10 of the *PCNT* gene, leading to a frameshift (Treo510fs) and premature protein truncation. The patient died because of a second fatal cerebrovascular accident.

**Patient 3** is a Caucasian female born prematurely with unknown familiar history. Birth weight was −1 SDS, birth length −1.99 SDS, head circumference −2.28 SDS. At birth, dysmorphic features were present with a high forehead with receded hairline, sparse scalp hair, and retrognathia. A more accurate clinical evaluation at one year revealed mild psychomotor delay, microcephaly, short neck, lower limbs hypertonia with hyperreflexia, atrial septal defects and patent foramen ovale (PFO), skeletal maturation delay tendency, metaphyseal widening, and delay in the femoral head ossification nuclei.

Considering the clinical spectrum, we proposed the diagnosis of MOPDII syndrome, which was confirmed by the absence of the pericentrin signal in the Western blot analysis (Figure 1). The subsequent evaluation of the *PCNT* gene did not identify any pathogenic variant.

### 4.2. Comparative Genomic Hybridization Array

Genomic DNA was extracted from the blood sample. Array-CGH analysis was performed using the Cytochip oligo ISCA 4 × 180K (Techno Genetics Srl, Avellino, Italy) following the manufacturer’s protocol.

### 4.3. DNA and RNA Extraction

Total DNA was extracted from the blood of the three patients and the healthy parents of Patient 1 using Eurogold blood DNA mini kit (Euroclone, Pero Milano, Italy). Total RNA was extracted from the blood of the three patients and age- and sex-matched control children. The blood was collected using specific “BD Vacutainer Safetyl-lok Blood Collection” and total RNA was extracted using a “Paxgene Blood RNA” Kit (Qiagen, Hilden, Germany).

### 4.4. Exome and Transcriptome Sequencing

DNA libraries were constructed using the “SureSelect QXT Human All Exon V5 + UTR” (Agilent Technologies, Santa Clara, CA, USA) protocol and sequenced on HiSeq Illumina platform, following the protocol (Illumina, San Diego, CA, USA).

The cDNA libraries were prepared using a platform-independent RNA-Seq protocol developed in our laboratories and sequenced using a 454 FLX Roche platform (Roche, Basilea, Swiss) [31].

### 4.5. Reverse Transcription and Real Time PCR Analysis

A quantity of 200 ng of total RNA was retrotranscribed using QuantiTect^®^ Reverse Transcription kit (Qiagen^®^ Hilden, Germany, according to the manufacturer’s instruction. Real-time PCR reactions were performed on Applied Biosystems™ 7900HT (Waltham, MA, USA). The glyceraldeyde 3P-dehydrogenase (GAPDH) was used as housekeeping gene because it has been listed by Normfinder as the most stable in samples used in qRT-PCR experiments.

To control the statistical significance of differentially gene expression levels, two-tailed Student’s t test was performed. The reported data represent the average of at least three independent experiments and are shown with their standard error (* *p* value < 0.05; ** *p* value < 0.005; *** *p* value < 0.0005).

### 4.6. Bioinformatics and Statistical Analyses

#### 4.6.1. Exome Analyses

Variant calling was performed according to the CoVaCS workflow [32] on hg19 reference assembly of the human genome. To reduce possible false positive calls, only variants supported by at least 10 independent reads were considered. Similarly, variants associated with genomic regions of low “mappability” (below 0.25 according to the GEM tool) [33] or with inconsistent mapping between the hg19 and hg38 human reference genome were discarded. Functional annotation of genes was performed by Annovar [34]. The following annotation resources were considered for the estimation of allele frequencies: ExA [35], 1000 Genomes (The 1000 Genomes Project Consortium, 2015) (phase 3) (version 2.1, updated 10 December 2018), dbSNP [36] (build 151), (version 160204-Public). RefSeq release 106 [37] was used for gene and transcript annotations, ClinVar [38] (version 1.55, updated 26 December 2018) and HGMD-Pro 2018.324 [39] for the annotation of disease-causing variants, and the dbNSPF [40] (v4.0b1, updated 30 December 2018) database for the evaluation of nonsynonymous substitutions effect. The identified nucleotides alterations were described based on Human Genome Variation Society nomenclature criteria (https://varnomen.hgvs.org/, (accessed on 12 October 2022)). The clinical classification of the variants was carried out according to the American College of Medical Genetics and Genomics (ACMG) criteria.

#### 4.6.2. Post-Processing of Exome Data and Variant Prioritization

Coordinates for the target regions on the hg19 human genome reference assembly, for the Agilent SureSelect QXT Human All Exon V5 + UTR kit, were obtained from the manufacturer’s web site. Coverage profiles were established using the bedtools utility [41]. Variant filtration was performed through a custom Perl script, and subsequently refined by expert manual curation. To identify potentially disease-causing variants, we considered only variants with a predicted disruptive effect (frameshift, splice site variants, stop-gain, and stop-loss variants, or CADD score > 20 for nonsynonymous variants), a minor allele frequency ≤ 1 × 10^−5^ in any of the resources of human genetic variation considered in the study, and homozygous in the affected individuals, but not in the healthy parents of Patient 1. Compound heterozygous variants, and candidate de novo variants, were identified by means of a custom Perl script. Since genetic profiles of the parents where only available for Patient 1, for Patient 2 and Patient 3 candidate compound heterozygous variants were identified as distinct heterozygous variants with a minor allele frequency ≤ 1 × 10^−5^, associated with the same gene and predicted to have a deleterious effect on the gene function according to the same criteria outlined above.

#### 4.6.3. RNA-Seq Analyses

Transcriptome data obtained with the Roche-454 sequencing platform were analyzed with the following steps. Data was mapped with BLAST against the Ensembl Human Transcript database, (Release-95) and read counts were performed with MultiDEA [42]. Differential expression analysis on these data was computed with Fold Change, Fisher’s Exact test (adjusted with False Discovery Rate), and with MultiDEA. Changes in gene expression were considered significant if absolute log_2_ Fold Change (log2FC) ≥ 0.585 (i.e., absolute Fold Change ≥ 1.5) and adjusted *p*-value ≤ 0.05. Pathway analysis of differentially expressed genes was performed with tools in The Database for Annotation, Visualization and Integrated Discovery (DAVID) v6.8 [43] and GeneCard’s tool GenesLikeMe, Version 3.12.404 [44].

#### 4.6.4. Western Blot Analysis

The levels of expression of pericentrin were determined by Western blot analysis using a rabbit pericentrin polyclonal antibody (ab4448, Abcam Ltd., Cambridge, UK) and the beta actin (A2066, Sigma-Aldrich, Inc., St. Louis, MO, USA) for normalization. The cells washed with PBS buffer plus 0.1 mM Na3VO4 were pelleted and lysed in Laemmli buffer (0.125 M Tris–HCl pH 6.8, 5% SDS) containing protease inhibitors. Lysates were boiled for 2 min, sonicated, and quantitated by Bradford assay. Aliquots containing 30 mg/mL of protein plus 5% b-mercaptoethanol were size fractionated on 3–8% Tris–acetate gel, using the NuPage Novex system from Invitrogen (Carlsbad, CA, USA). After incubation with a peroxidase-conjugated secondary antibody, the immunoreactive bands were visualized by ECL Supersignal on autoradiographic films.

## Figures and Tables

**Figure 1 ijms-24-12291-f001:**
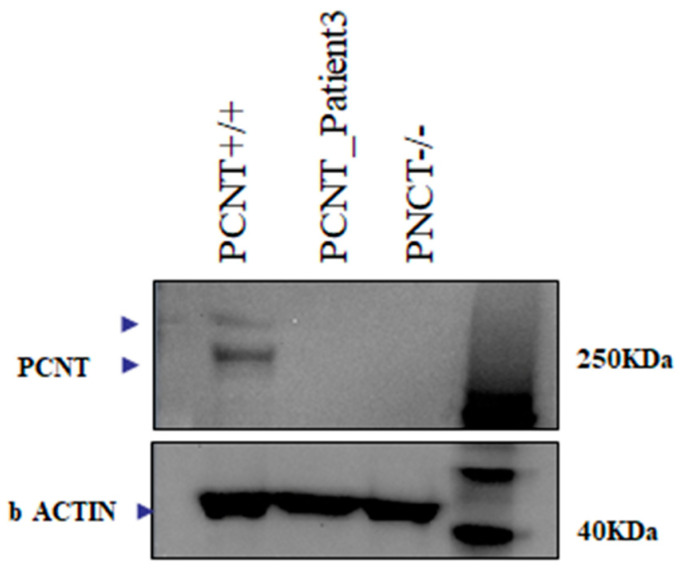
PCNT protein expression in MOPDII Patient 3. Western blotting analysis of pericentrin in cellular extracts from lymphoblastoid cell lines (LCLs) from PCNT wild-type control (lane 1); MOPD II patient 3 (lane 2); PCNT homozygous for c.1523dup (lane 3).

**Figure 2 ijms-24-12291-f002:**
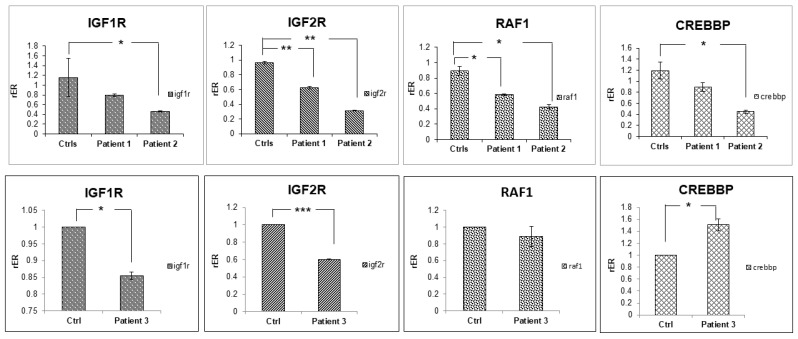
RT-qPCR of IGF1R, IGF2R, RAF, and CREBBP in the three MOPDII patients. For Patients 1 and 2, the mean expression of the indicated genes in three age-matched male children was used as calibrator. For Patient 3, the mean expression of the indicated genes in three age-matched female children was used as calibrator. Data are shown as the average with a standard error of three independent experiments (* *p* value < 0.05; ** *p* value < 0.005; *** *p* value < 0.0005).

**Table 1 ijms-24-12291-t001:** Disease-causing variants with a minor allele frequency (MAF) ≤ 1 × 10^−5^. ACMG Criteria Used for Classification of Pathogenicity score: C3 = Variant of Uncertain Significance, C4 = Likely Pathogenic; C5 = Pathogenic. * = potentially deleterious compound heterozygous variants.

Case	Refseq	Gene	Genotype	Exon/ Intron	cDNA (HGVS)	Protein Change (HGVS)	Mutation Type	Gnom AD	dbSNP	ACMG	Class	Gene Function
1	NM_006031.6	*PCNT*	Hmz	Intron 18	c.3608-2A>G	-	splicing	-	-	PVS1, PM2	C4	The protein encoded by this gene binds to calmodulin and is expressed in the centrosome. It is an integral component of the pericentriolar material (PCM). The protein interacts with the microtubule nucleation component gamma-tubulin and is likely important to normal functioning of the centrosomes, cytoskeleton, and cell-cycle progression. MutationPathogenic variants in this gene cause Seckel syndrome-4 and microcephalic osteodysplastic primordial dwarfism type II
2	NM_006031.6	*PCNT*	Hmz	Exon 10	c.1523dupA	p.(Thr510AsnfsTer4)	indel	0.0000319	rs1369869782	PVS1, PM2, PP5	C5	
	NM_031419.4	*NFKBIZ*	Hmz	Exon 11	c.1635+2-ACTTTTAGAA	-	splicing	0.0000489	-	PP3, PM2	C3	This gene is a member of the ankyrin-repeat family and is induced by lipopolysaccharide (LPS). The C-terminal portion of the encoded product which contains the ankyrin repeats, shares high sequence similarity with the I kappa B family of proteins. The latter are known to play a role in inflammatory responses to LPS by their interaction with NF-B proteins through ankyrin-repeat domains. Studies in mouse indicate that this gene product is one of the nuclear I kappa B proteins and an activator of IL-6 production
3	NM_014739.3	*BCLAF1*	Hmz	Intron 10	c.2397+1G>C	-	splicing	-	-	PVS1, PM2	C4	This gene encodes a transcriptional repressor that interacts with several members of the BCL2 family of proteins. Overexpression of this protein induces apoptosis. The protein localizes to dot-like structures throughout the nucleus, and redistributes to a zone near the nuclear envelope. Diseases associated with BCLAF1 include Emery-Dreifuss Muscular Dystrophy and Uterine Adnexa Cancer. Among its related pathways are Interactome of polycomb repressive complex 2 (PRC2).
	NM_002723	*PRB4*	Hmz	Exon 3	c.363_364ins GACGACCC…	-	frameshift insertion	-	-	PVS1_Moderate + PM1 + PM2 + PP3	-	This gene encodes a member of the heterogeneous family of basic, proline-rich, human salivary glycoproteins. The encoded preproprotein undergoes proteolytic processing to generate one or more mature peptides before secretion from the parotid glands.
	NM_000552	*VWF*	Hmz	Exon 28	c.4165_4166ins ACCAGCGAGGTC…	-	stopgain	-	-	PVS1	-	This gene encodes a member of the heterogeneous family of basic, proline-rich, human salivary glycoproteins. The encoded preproprotein undergoes proteolytic processing to generate one or more mature peptides before secretion from the parotid glands.
2 *	NM_001077624.3	*ZNF846*	Het		c.885_886insGA	p.Tyr296AspfsTer63	Frameshift	0	-	PM2	3	This gene encodes a predicted protein to enable DNA-binding transcription repressor activity, RNA polymerase II-specific and RNA polymerase II transcription regulatory region sequence-specific DNA binding activity. Predicted to be involved in negative regulation of transcription by RNA polymerase II.
2 *	NM_001077624.3	*ZNF846*	Het		c.884C>A	Ser295Leu	Missense	0.000008	rs765545468	PM2	3	
3 *	M_001374736.1	*DST*	Het		c.9207A>T	Arg3069Ser	Missense	0	rs1364606135	PM2, BP1	3	This gene encodes a member of the plakin protein family of adhesion junction plaque proteins. Multiple alternatively spliced transcript variants encoding distinct isoforms have been found for this gene, but the full-length nature of some variants has not been defined. It has been reported that some isoforms are expressed in neural and muscle tissue, anchoring neural intermediate filaments to the actin cytoskeleton, and some isoforms are expressed in epithelial tissue, anchoring keratin-containing intermediate filaments to hemidesmosomes. Consistent with the expression, mice defective for this gene show skin blistering and neurodegeneration.
3 *	M_001374736.1	*DST*	Het		c.6371C>A	p.Ala2124Glu	Missense			PM2, BP1	3	

**Table 2 ijms-24-12291-t002:** Genes with two or more rare, deleterious variants in MOPDII patients.

Case	Gene	Function
1-2-3	RAD17	Cell cycle checkpoint protein. Essential for sustained cell growth, maintenance of chromosomal stability, and ATR-dependent checkpoint activation upon DNA damage. Has a weak ATPase activity required for binding to chromatin. May also serve as a sensor of DNA replication progression, and may be involved in homologous recombination.
2-3	NPRL3	As a component of the GATOR1 complex functions as an inhibitor of the amino acid-sensing branch of the TORC1 pathway. Important role in cortical development.
2-3	BCLAF1	Death-promoting transcriptional repressor. Bclaf1 Promotes Maintenance and Self-Renewal of Fetal Hematopoietic Stem Cells.
3-1	PRIM2	Regulatory subunit of the DNA primase complex and component of the DNA polymerase alpha complex (also known as the alpha DNA polymerase-primase complex) which play an essential role in the initiation of DNA synthesis.

## Data Availability

The data that support the findings will be available in a repository following publication, the authors are taking care.

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
