# Peer review of "Whole-Exome and Transcriptome Sequencing Expands the Genotype of Majewski Osteodysplastic Primordial Dwarfism Type II"

_ijms, 2023, doi:10.3390/ijms241512291_

Round 1
Reviewer 1 Report
Marzano et al. conducted analysis on three MOPDII patients. Among them, Patient 1 and 2 received molecular diagnosis as reported in previous published papers (homozygous LoF changes in the PCNT gene). Patient 3 received a clinical diagnosis of MOPDII based on her phenotype, yet no pathogenic variants in PCNT were identified; the authors claimed her diagnosis “was confirmed by the absence of the pericentrin signal in the Western Blot analysis”, although the Western blot result could neither be found in this manuscript nor with reference to any previous publication.
The aim of the study is to explain the complex phenotype of the MOPDII disease by identifying disease-causing variants beyond those in the PCNT gene. The authors performed exome sequencing on the three patients and RNA-seq on blood sample of Patient 1. There are several shortcomings intrinsic to the experimental design and analysis, as well as concerns on the validity of the conclusions drawn from the authors:
For exome sequencing analysis:
1. Coverage of the PCNT gene is quite low (Table S2, ~88% for 10x DP in Patient 3). This may be the reason why disease-causing variants in the PCNT gene was not identified in this individual.
2. It is unclear why only homozygous variants were considered (Table 1, Table 2). What about compound heterozygous and de novo heterozygous changes? For inherited heterozygous changes in patients, they could also be disease-causing with reduced penetrance.
3. It is also unclear why only LoF variant were considered (Table 1). Other variant types could also be disease causing, e.g. missense, inframe indel, splicing variant beyond +/- 2bp, etc.
3. None of the variants listed in Table 1 and 2 were described with proper curation; the authors should follow the ACMG guideline (PMID 25741868) and state the category of these variants to be pathogenic or not.
4. Is the third column in Table 2 (“MAX ALLELE FREQUENCY”) refers to the allele frequencies calculated from control databases like ExAc, 1000 Genomes or gnomAD? As these variants are extremely common in the unaffected populations, how to justify them causing a rare disease like MOPDII? If the authors meant to perform burden analysis, these variants are too common; if the authors meant to perform GWAS with common variants, the current analysis with three patients is obviously insufficient.
For RNA-seq analysis:
1. “Blood samples from Patient 1, and two healthy age and sex-matched controls” were subjected to transcriptome analysis. Since results obtained from RNA-seq are extremely variable, for clinical diagnosis purpose, data from hundreds (or at least dozens) of samples are necessary to establish a baseline expression level for each gene. Only one patient sample and two control samples are not sufficient to generate any reliable result.
2. Since MOPDII is a prenatally onset disease mainly affecting the skeletal system, the appropriateness of using postnatal blood as a sample type for transcriptomic study is questionable. Particularly, expression of the PCNT gene was not shown to be dysregulated (Table S4). Given the LoF canonical splicing site change in Patient 1, expression of PCNT should be absent or at least reduced; yet this was not reflected in the RNA-seq data, indicating the inaccuracy of these results.
3. For Patient 3 without PCNT variants identified, it may be helpful to study mRNA of the gene to identify potential intronic variants (or synonymous variants) causing alternative splicing, or copy-number variants/structural variations missed by exome.
4. There is no correlation analysis between the genomic DNA changes and the mRNA changes identified, although this should be done after proper experiments performed for exome sequencing and RNA-seq.
Some minor comments:
1. The genetics field has stopped using the terminology of “mutation”; pathogenic or disease-causing variants should be used instead.
2. Similarly, the field agreed that “whole exome sequencing” should be changed into clinical exome sequencing or exome sequencing, as it does not cover all the coding regions.
Typos and grammar errors could be spotted.
Author Response
We thank the reviewer for the insightful comments and excellent suggestions. We have carefully evaluated them, performed additional experiments and analyses (showed in the new tables of the revised paper) to address the points raised and fulfill the requests.
All the revisions in the manuscript are in red. Below our answers and clarifications point-by-point.
- “Marzano et al. conducted analysis on three MOPDII patients. Among them, Patient 1 and 2 received molecular diagnosis as reported in previous published papers (homozygous LoF changes in the PCNT gene). Patient 3 received a clinical diagnosis of MOPDII based on her phenotype, yet no pathogenic variants in PCNT were identified; the authors claimed her diagnosis “was confirmed by the absence of the pericentrin signal in the Western Blot analysis”, although the Western blot result could neither be found in this manuscript nor with reference to any previous publication.”
We fully agree with the reviewer, we performed the Western blot experiment (see Figure 2) in the new version of the paper).
For exome sequencing analysis:
- Coverage of the PCNT gene is quite low (Table S2, ~88% for 10x DP in Patient 3). This may be the reason why disease-causing variants in the PCNT gene was not identified in this individual.
We agree with the referee on this point, anyway our observations with respect to the absence of potentially disease-causing variants in Patient 3, based on WES alone cannot be considered as conclusive. Importantly, targeted Sanger re-sequencing of PCNT exons was performed and confirmed the WES analysis. We included this result in the new version of the paper (see lines 127-129).
- It is unclear why only homozygous variants were considered (Table 1, Table 2). What about compound heterozygous and de novo heterozygous changes? For inherited heterozygous changes in patients, they could also be disease-causing with reduced penetrance.
We thank the referee for this suggestion. In the new version of the paper, we evaluated also potential de-novo and compound heterozygous variants only in patient 1, as we only had the parents of patient 1 available. A similar consideration applies also for the compound heterozygous variants: in the absence of the parent’s genotypes patient 2 and patient 3 variants can not be phased/assigned to a haplotype.
As a proxy, genes with two heterozygous, rare, and deleterious variants were evaluated (see potential compound heterozygous in table 1).
By performing these analyses, 2 additional genes with potentially compound heterozygous, rare, deleterious variants were identified: ZNF846 in patient 2; DST in patient 3.
Importantly, disease-causing variants were already identified in patient 2 and patient 1 by our variant prioritization strategy. DST is implicated in Epidermolysis Bullosa Simplex 3 and Neuropathy, Hereditary Sensory and Autonomic, Type Vi (HSAN6).
- It is also unclear why only LoF variant were considered (Table 1). Other variant types could also be disease causing, g. missense, inframe indel, splicing variant beyond +/- 2bp, etc.
As rightly suggested by the reviewer, inframe indel, and splicing variants were not excluded by our variant prioritization workflow. Indeed, table 1 features two frameshift-insertion variants and two splicing variants. For example the disease-causing variant for patient1 is a splicing variant. We do now evaluate missense variants. Missense variants reported as "deleterious" by two independent methods for the evaluation of the effects of missense variants (SIFT and CADD ) and with an allele frequency lower than the threshold are included in table 1. The main text was edited to clarify this point.
- None of the variants listed in Table 1 and 2 were described with proper curation; the authors should follow the ACMG guideline (PMID 25741868) and state the category of these variants to be pathogenic or not.
We thank referee1 for this suggestion. Variants reported in Table 1 are now annotated according to ACMG guidelines as requested. Table 2 was completely overhauled, it now reports genes rather than variants.
- Is the third column in Table 2 (“MAX ALLELE FREQUENCY”) refers to the allele frequencies calculated from control databases like ExAc, 1000 Genomes or gnomAD? As these variants are extremely common in the unaffected populations, how to justify them causing a rare disease like MOPDII? If the authors meant to perform burden analysis, these variants are too common; if the authors meant to perform GWAS with common variants, the current analysis with three patients is obviously insufficient.
We agree with the referee on this point. Table 2, as in its previous form, was removed. The current table 2 reports a list of genes associated with rare, deleterious variants (homozygous or heterozygous) in one or more patients.
For RNA-seq analysis:
- “Blood samples from Patient 1, and two healthy age and sex-matched controls” were subjected to transcriptome analysis. Since results obtained from RNA-seq are extremely variable, for clinical diagnosis purpose, data from hundreds (or at least dozens) of samples are necessary to establish a baseline expression level for each gene. Only one patient sample and two control samples are not sufficient to generate any reliable result.
We thank the reviewer for this consideration, but MOPDII is a rare pathology, 150 individuals in the word with molecularly confirmed MOPDII have been identified. Therefore 3 subjects with Majewski's syndrome is already a "high" number. However, we confirm RNA-seq results also using RT-qPCR.
- Since MOPDII is a prenatally onset disease mainly affecting the skeletal system, the appropriateness of using postnatal blood as a sample type for transcriptomic study is questionable. Particularly, expression of the PCNT gene was not shown to be dysregulated (Table S4). Given the LoF canonical splicing site change in Patient 1, expression of PCNT should be absent or at least reduced; yet this was not reflected in the RNA-seq data, indicating the inaccuracy of these results.
We thank the reviewer for this consideration, but the pathology is diagnosed after birth, a prenatal investigation would not have been possible. In addition to the RNA-Seq analysis we performed RT-qPCR. The expression of the PCNT transcript remains unchanged in Patient 1, confirming the RNA-Seq results. What happens in patient 1, due to the mutation in the splicing site, is the formation of a transcript of different size than the control but equally expressed. Compared to a healthy subject, in patient 1 the mutation determines the loss of exon 19. To verify this, we amplified and sequenced the PCNT cDNA (see Figure B in ppt). As already mentioned by colleagues in a previous work (Willems et al., J Med Genet. 2010), the mutation impairs exon 19 splicing, leading to the formation of premature STOP codon.
- For Patient 3 without PCNT variants identified, it may be helpful to study mRNA of the gene to identify potential intronic variants (or synonymous variants) causing alternative splicing, or copy-number variants/structural variations missed by exome.
We thank the reviewer for this consideration. In patient 3 the lack of the protein despite wild-type PCNT exonic sequence and a transcript level comparable to control (see Figure C in ppt) could be due to different causes as the alteration of post-transcriptional mechanisms as specific miRNAs upregulation. Therefore this point needs further investigations which are not the focus of this paper.
- There is no correlation analysis between the genomic DNA changes and the mRNA changes identified, although this should be done after proper experiments performed for exome sequencing and RNA-seq.
We thank the reviewer for this consideration, but there is often no direct correlation between exonic mutations and gene expression. Genomic variations may not alter gene expression, or may cause alterations in post-transcriptional regulation.
Some minor comments:
- The genetics field has stopped using the terminology of “mutation”; pathogenic or disease-causing variants should be used instead.
- Similarly, the field agreed that “whole exome sequencing” should be changed into clinical exome sequencing or exome sequencing, as it does not cover all the coding regions.
We thank the reviewer for these considerations, we have modified the text as suggested.

Reviewer 2 Report
The manuscript focuses on a very rare disease, MOPDII, whose characteristic aspect is a complex phenotype currently only associated with pathogenic variants in one gene, PCNT. The authors undertook two techniques to detect if other genetic factors may contribute to the multiple clinical features of MOPDII. This aim is relevant since often the concept of Mendelian disease, one gene one phenotype, does not fully explain the expression of the disease. I positively judge the research design, especially about the criteria adopted in the data analysis. Nevertheless, I ask attention on some points:
- line 84 “A total of 171.352 genetic variants were identified in the exomes of the five individuals included in our study”: here I suggest to repeat that these are the three patients and the two parents of one of them;
- line 124-127: here I suggest explaining better this problem with the WES;
- line 177- 180: I think these results merit attention and request further explanation;
- line 217-218, about patient 3: I suggest adding here if this case was also analysed with the duplication-deletion analysis method to detect any structural variants involving the PCNT gene or I suggest adding if other type of analysis are in progress;
- line 327: as suggested above, I think it is important to underline that for patient 3 the definition of the PCNT genotype is extremely relevant because it would confirm the diagnosis of MOPDII (line 207-208) and allow for a more appropriate family counselling
Author Response
We really thank the reviewer for the insightful comments and excellent suggestions. We have carefully evaluated them, performed additional experiments and analyses (showed in the new figures of the revised paper) to address the points raised and fulfill the requests.
All the revisions in the manuscript are in red. Below our answers and clarifications point-by-point
Nevertheless, I ask attention on some points:
- line 84 “A total of 171.352 genetic variants were identified in the exomes of the five individuals included in our study”: here I suggest to repeat that these are the three patients and the two parents of one of them;
We thank the reviewer for this consideration, we have added the suggested phrase.
- line 124-127: here I suggest explaining better this problem with the WES;
We thank the reviewer for this suggestion. For patient 3 the results obtained with ES cannot be conclusive because we do not see enough coverage. The variants could be in 14% of the gene for which we have little data, so we could lose some variants. Importantly, targeted Sanger re-sequencing of PCNT exons was performed and confirmed the WES analysis.
- line 177- 180: I think these results merit attention and request further explanation;
We thank the reviewer for this consideration, we have eliminated this part from the new version of the paper.
- line 217-218, about patient 3: I suggest adding here if this case was also analysed with the duplication-deletion analysis method to detect any structural variants involving the PCNT gene or I suggest adding if other type of analysis are in progress;
We thank the reviewer for this consideration. The causes that determine the presence of a wild type gene but with an absent protein can be different. There may be duplications, deletions or mutations in introns that are only visible with a whole genome sequencing approach. Moreover, the normally expressed transcript may not be functional due to the alteration of post-transcriptional mechanisms or due to the role played by miRNAs that prevent the production of the protein. However, the possible regulatory mechanisms will be investigated with in-depth analyzes which are the subject of a future paper.
- line 327: as suggested above, I think it is important to underline that for patient 3 the definition of the PCNT genotype is extremely relevant because it would confirm the diagnosis of MOPDII (line 207-208) and allow for a more appropriate family counselling. We thank the reviewer for this consideration. For Patient 3 we did not identify pathogenic variants in PCNT gene. Targeted Sanger re-sequencing of PCNT exons in the same patient confirm this result. The absence of PCNT protein needs further investigations which will be the focus of future work.

Round 2
Reviewer 1 Report
The revised manuscript addressed most of this reviewers’ previous concerns, yet raising new ones based on additional data. Western blot in Figure 2 indicates no PCNT expression in Patient 3, yet qPCR result in Figure C demonstrates equivalent mRNA level of the gene as the control. The authors claim this “could be due to different causes as the alteration of post-transcriptional mechanisms as specific miRNAs upregulation. Therefore this point needs further investigations which are not the focus of this paper”. Post transcriptional regulation, particularly the miRNAs mentioned by the authors, may decrease its expression level, but hardly to a non-existing status. Therefore, the reliability of the qPCR result in Figure C is questionable. To clarify this, please use multiple qPCR primers covering different exon-exon junctions to detect mRNA level of the gene. Since this manuscript involves transcriptomic analysis with RNA-seq (yet done in only one sample), while most of the mRNA levels are illustrated by qPCR, to demonstrate the reliability of your qCPR method is critical.
Further, the authors claim “the expression of the PCNT transcript remains unchanged in Patient 1, confirming the RNA-Seq results”. Statistical analysis is missing in Figure C; the fold change of patient 1 comparing to the control is 0.7 and seems to be significantly different from the control. To compare, IGF1R qPCR result in Figure 1 shows 0.85 fold change yet with statistical significance. Please repeat the qPCR in patent 1 as suggested above and perform statistical analysis. This is critical to demonstrate your RNA-seq result is dependable and consistent with the qPCR validation.
Figure and Table legends are missing or incomplete, e.g. unclear what is source of the negative control PNCT-/- sample in Figure 2, and in Table 1, what are * signs labelling and what are C3/C4/C5/3 under column “Class”.
SUBJECTS AND METHODS section is not updated according to the edits in the revised manuscript, e.g. no Western blot method included.
Typos and grammar errors could be spotted.
Author Response
Dear Editor,
Thank you for considering our manuscript (Manuscript ID ijms-2396393): “WHOLE-EXOME AND TRANSCRIPTOME SEQUENCING EXPANDS THE GENOTYPE OF MAJEWSKI OSTEODYSPLASTIC PRIMORDIAL DWARFISM TYPE II” by Flaviana Marzano et al. for publication in International Journal of Molecular Sciences - Molecular Genetics and Genomics.
We sincerely thank you and the reviewers for the constructive criticisms and valuable comments, which were of great help in revising the manuscript.
We have carefully evaluated the reviewer’s comments and performed the requested analyses to address their points and fulfill their remarks.
The revised version of the manuscript contains all the revisions required with new figures.
All the revisions in the manuscript are highlighted in red, moreover reviewers’ responses have been supported by additional figures not included in the manuscript .
Your Sincerely.
Reviewer 1
Answer
We thank the reviewer for the insightful comments and excellent suggestions. We have carefully evaluated them, performed additional analyses to address the points raised and fulfill the requests.
All the revisions in the manuscript are in red. Below our answers and clarifications point-by-point.
- “The revised manuscript addressed most of this reviewers’ previous concerns, yet raising new ones based on additional data. Western blot in Figure 2 indicates no PCNT expression in Patient 3, yet qPCR result in Figure C demonstrates equivalent mRNA level of the gene as the control. The authors claim this “could be due to different causes as the alteration of post-transcriptional mechanisms as specific miRNAs upregulation. Therefore this point needs further investigations which are not the focus of this paper”. Post transcriptional regulation, particularly the miRNAs mentioned by the authors, may decrease its expression level, but hardly to a non-existing status. Therefore, the reliability of the qPCR result in Figure C is questionable. To clarify this, please use multiple qPCR primers covering different exon-exon junctions to detect mRNA level of the gene. Since this manuscript involves transcriptomic analysis with RNA-seq (yet done in only one sample), while most of the mRNA levels are illustrated by qPCR, to demonstrate the reliability of your qPCR method is critical.
We fully agree with the reviewer, but when we claim this “could be due to different causes as the alteration of post-transcriptional mechanisms as specific miRNAs upregulation. Therefore this point needs further investigations which are not the focus of this paper”, we were referring that there could be a miRNA-mediated mechanism of inhibition of translation. We agree with the reviewer that miRNA could give a reduction and not a total absence of the mRNA, for this reason we stated that this case needs further investigations. The reviewer's request for multiple qPCR primers covering different exon-exon junctions to detect mRNA level of the gene is very complex. The PCNT gene has 47 exons so this would mean creating, validating and using 47 pairs of primers, impossible to do in the required time of ten days. If the reviewer requests remains, we could perform an RNA-seq on patient 3 and check the PCNT transcripts. For such a study 10 days would not be sufficient.
In a future study we could perform native RNA sequencing using Nanopore platform, in order to sequence the entire mature RNA.
- Further, the authors claim “the expression of the PCNT transcript remains unchanged in Patient 1, confirming the RNA-Seq results”. Statistical analysis is missing in Figure C; the fold change of patient 1 comparing to the control is 0.7 and seems to be significantly different from the control. To compare, IGF1R qPCR result in Figure 1 shows 0.85 fold change yet with statistical significance. Please repeat the qPCR in patent 1 as suggested above and perform statistical analysis. This is critical to demonstrate your RNA-seq result is dependable and consistent with the qPCR validation.
We thank the referee for this suggestion, RT-qPCR for PCNTs represent the mean of at least three experiments. As suggested, we have added statistical analysis (see new version of Figure C attached to these replies). The difference in expression is statistically significant only for patient 2 (p-value ≤0.05). Moreover, we report in Figure C the normalized data obtained from RNA-seq, this result (fold change=0.7) is confirmed by RT-qPCR. In the PCNT amplification a fold change of 0.7 is non-statistically significant because the PCNT expression in the controls is more variable than IGF1R expression.
- Figure and Table legends are missing or incomplete, e.g. unclear what is source of the negative control PNCT-/- sample in Figure 2, and in Table 1, what are * signs labelling and what are C3/C4/C5/3 under column “Class”.
We thank the referee for this suggestion. In the new version of the paper, we add Figure Legend 2. Western blotting analysis of pericentrin in cellular extracts from lymphoblastoid cell lines (LCLs) from PCNT wild type control (lane 1); MOPD II patient 3 (lane 2); PCNT homozygous for c.1523dup (lane3).
Table Legend: ACMG Criteria Used for Classification of Pathogenicity score: C3=Variant of Uncertain Significance, C4=Likely Pathogenic; C5=Pathogenic
- SUBJECTS AND METHODS section is not updated according to the edits in the revised manuscript, e.g. no Western blot method included.
As rightly suggested by the reviewer, we add to Methods these parts:
In exome analysis: The identified nucleotides alterations were described based on Human Genome Variation Society nomenclature criteria (https://varnomen.hgvs.org/). The clinical classification of the variants was carried out according to the American College of Medical Genetics and Genomics (ACMG) criteria.
Western Blot analysis: The levels of expression of pericentrin were determined by western blot analysis using a rabbit pericentrin polyclonal antibody (ab4448, Abcam Ltd, Cambridge, UK) and the beta actin (A2066, Sigma-Aldrich, Inc., St. Louis, MO) for normalization. The cells washed with PBS buffer plus 0.1 mM Na3VO4 were pelleted and lysed in Laemmli buffer (0.125 M Tris–HCl pH 6.8, 5% SDS) containing protease inhibitors. Lysates were boiled for 2 min, sonicated and quantitated by Bradford assay. Aliquots containing 30 mg/ml of protein plus 5% b-mercaptoethanol were size fractionated on 3–8% Tris–acetate gel, using the NuPage Novex system from Invitrogen (Carlsbad, CA). After incubation with a peroxidase-conjugated secondary antibody the immunoreactive bands were visualized by ECL Supersignal on autoradiographic films.

Round 3
Reviewer 1 Report
1. Regarding repeating the qPCR analysis of PCNT in patients, no need to design 47 pairs of primers to cover all exon-exon junctions, but at least two additional primer pairs should be added to cover different regions of the gene, besides the current primer pair used; these primer pairs should show consistent fold change results comparing to the control. This reviewer was not aware that only 10 days were given for the manuscript revision. Longer period of time should be allowed.
2. After repeating the qPCR, analysis results of PCNT should be included in the manuscript.
3. Include sequences of all the primers used in this study as a supplementary table.
N/A
Author Response
Dear Editor,
Thank you for considering our manuscript (Manuscript ID ijms-2396393): “WHOLE-EXOME AND TRANSCRIPTOME SEQUENCING EXPANDS THE GENOTYPE OF MAJEWSKI OSTEODYSPLASTIC PRIMORDIAL DWARFISM TYPE II” by Flaviana Marzano et al. for publication in International Journal of Molecular Sciences - Molecular Genetics and Genomics.
We sincerely thank you and the reviewers for the constructive criticisms and valuable comments, which were of great help in revising the manuscript.
We have carefully evaluated the reviewer’s comments and performed the requested experiments and analyses to address their points and fulfill their remarks.
All the revisions in the manuscript are highlighted in red, moreover reviewers’ responses have been supported by additional figures not included in the manuscript .
Your Sincerely.
Reviewer 1
Answer
We thank the reviewer for the insightful comments and excellent suggestions. We have carefully evaluated them, performed additional analyses to address the points raised and fulfill the requests.
All the revisions in the manuscript are in red. Below our answers and clarifications point-by-point.
- Regarding repeating the qPCR analysis of PCNT in patients, no need to design 47 pairs of primers to cover all exon-exon junctions, but at least two additional primer pairs should be added to cover different regions of the gene, besides the current primer pair used; these primer pairs should show consistent fold change results comparing to the control. This reviewer was not aware that only 10 days were given for the manuscript revision. Longer period of time should be allowed.
- After repeating the qPCR, analysis results of PCNT should be included in the manuscript.
- Include sequences of all the primers used in this study as a supplementary table.
To evaluate if in Patient 3 the PCNT gene is entirely transcribed we performed an RNA-seq experiment. The PCNT transcript reads cover a high number of exons (29 on 48 exons) and importantly all the last exons, thus indicating that the entire gene is transcribed (see figure below).
We can conclude that the absence of protein in this patient could be due to the alteration of post-transcriptional mechanisms that need further investigations that go beyond the focus of this paper.
We added this sentence into the text “The RNA-seq analysis show that PCNT transcript reads cover a high number of exons (29 on 48 exons) and importantly all the last exons, thus indicating that the entire gene is transcribed (data not shown)”.

Round 4
Reviewer 1 Report
N/A
N/A